# Numerical Simulation of Sediment Transport in Unsteady Open Channel Flow

Jennifer G. Duan [1,*], Chunshui Yu [1] and Yan Ding [2]

1 Department of Civil and Architectural Engineering and Mechanics, University of Arizona, Tucson, AZ 85716, USA; chunshui@email.arizona.edu
2 Research Civil Engineer (Hydraulics), U.S. Army Engineer Research and Development Center (ERDC), Coastal and Hydraulics Laboratory (CHL), Vicksburg, MS 39180, USA; yan.ding@erdc.dren.mil
* Correspondence: gduan@arizona.edu

**Abstract:** This paper presented a two-dimensional, well-balanced hydrodynamic and sediment transport model based on the solutions of variable-density shallow-water equations (VDSWEs) and the Exner equation for bed change for simulating sediment transport in unsteady flows. Those equations are solved in a coupled way by the first-order Godunov-type finite volume method. The Harten–Lax–van Leer–Contact (HLLC) Riemann solver is extended to find the local Riemann fluxes to maintain the exact balance between the momentum term and the bed slope term. A well-balanced scheme is superior to an unbalanced scheme to minimize numerical dispersion as demonstrated by the synthetic standing contact-discontinuity test case. Following this, the model is employed to simulate two laboratory experiments and a field case, the 1996 Lake Ha! Ha! flood event in Canada. The results of water surface elevations and bed surface profiles agree well with the measurements. The accuracy and robustness of the numerical schemes make the model a good candidate for practical engineering applications.

**Keywords:** dam-break flow; Godunov-type finite volume method; HLLC Riemann solver; nonequilibrium sediment transport model; variable density shallow water equations; well-balanced property

## 1. Introduction

For unsteady open-channel flows over mobile beds, such as during a dam break, decommissioning, or levee or barrier breaching in a flood or storm surge, the spatial variation in flow densities due to sediment is considerable and may impose significant impacts on flood flows. The consequences include erosion and sedimentation on bed and banks, river platform evolution (e.g., avulsion, cut off), and failures of instream structures (e.g., levees, dams, bridges) (e.g., [1,2]). Since flow densities are not constants but change spatially, sediment transport and morphological changes should be modeled using variable-density shallow-water equations (VDSWEs) [1,3–12]. As the name implies, VDSWEs are an extension of shallow-water equations (SWEs) and are also based on the shallow-water assumption [13]. Besides this, two simplifications are employed in the derivation of the VDSWEs [1,14]: (1) the flow is dominated by suspended sediment load; (2) there are no obvious lags between sediment particles and water flow.

To account for their hyperbolic nature, VDSWEs are often solved by the Godunov-type finite volume method [6,15]. A thorough review of the application of the method to the numerical solutions of SWEs is given by [16]. In contrast with SWEs, the density of sediment-laden flow in the VDSWEs is an independent variable. To simplify the solution of the VDSWEs, refs. [1,17] reformulated the VDSWEs by moving the density-related terms to the right-hand side of the equations. After this reformulation, the left-hand side of the equations has the same form as the SWEs, while the right-hand side has two more terms than the SWEs. Ref. [1] pointed out that one additional term represents the effect of spatial variation in flow densities and the other one described the momentum exchange between

flow and mobile bed. Since the reformulated VDSWEs have the same left-hand side as the SWEs, it is a common practice to apply the previously well-developed Godunov-type SWE solvers to solve the VDSWEs and to discretize the additional terms by a central difference scheme [1,5,7,9].

However, one of the two additional terms on the right-hand side involves the gradient of density derived from the concentration of sediment. Mathematically, this term represents a product of the Heaviside function by the Dirac function [18]. Across discontinuous fronts of a density field, its mathematical meaning is not well-defined [19,20]. Numerically, this term originates from the advection term of the VDSWEs and it should be discretized by an upwind scheme. Ref. [21] argued that it is difficult to maintain the conservative property (C-property) [22] of numerical models based on the reformulated VDSWEs. Ref. [21] also suggested that the better way to create a well-balanced model is to solve the original formulation of the VDSWEs. This will minimize the errors of finite difference schemes used for discretization in the discontinuous contacts of two adjacent cells.

This paper presents the first two-dimensional model that solves the VDSWEs using a novel well-balanced numerical scheme for simulating unsteady sediment-laden flows (e.g., a dam break) over a mobile bed. The model is based on the original formulation of the VDSWEs. The terms derived from the gradient of sediment concentration are incorporated into the advective terms in the momentum equations. In the model, the Godunov-type finite volume method is applied to solve the system of governing equations simultaneously.

The rest of the paper is organized as follows: in Section 2, the governing equations, i.e., the two-dimensional variable-density shallow-water equations, and their two formulations, are presented. In Section 2.2, a well-balanced numerical scheme is developed for the VDSWEs. In Section 3, the proposed model is tested against four test cases and the results are discussed and compared with the available observations. Finally, the main conclusions are drawn in Section 4.

## 2. Governing Equations

In this study, the dam-break flows over mobile beds are described by the variable-density shallow-water equations (VDSWEs). In the VDSWEs, the mixture of water and sediment is treated as a continuum, and both phases are moving together at the same velocity. The system of governing equations consists of volume-, mass-, and momentum-conservation equations for flow and sediment. For a sediment-laden flow, the bulk volume-conservation equation is written as:

$$\frac{\partial h}{\partial t} + \frac{\partial (hu)}{\partial x} + \frac{\partial (hv)}{\partial y} = S_b \tag{1}$$

where $t$ is time; $x$ and $y$ are the spatial coordinates; $h$ is flow depth; $u$ and $v$ are the depth-averaged flow velocities in $x$ and $y$ directions, respectively; $S_b$ is the sediment exchange rate between flow and bed. Since the density of sediment-laden flow is a spatial function, the mass conservation equation of the sediment–water mixture is expressed as:

$$\frac{\partial (\rho h)}{\partial t} + \frac{\partial (\rho hu)}{\partial x} + \frac{\partial (\rho hv)}{\partial y} = \rho_b S_b \tag{2}$$

where $\rho$ is the density of sediment-laden flow; $\rho_b = \phi \rho_w + (1 - \phi)\rho_s$ is the density of the saturated mobile bed; $\phi$ is the porosity of the mobile bed; $\rho_w$ is the density of clear water; $\rho_s$ is the density of sediment. The concentration of sediment-laden flow can be calculated by $C = \frac{\rho - \rho_w}{\rho_s - \rho_w}$. The depth-averaged momentum-conservation equations in $x$ and $y$ directions are given as:

$$\frac{\partial (\rho hu)}{\partial t} + \frac{\partial}{\partial x}\left(\rho huu + \frac{1}{2}\rho g h^2\right) + \frac{\partial (\rho huv)}{\partial y} = \rho g h S_{0x} - \rho C_f \|\boldsymbol{u}\| u \tag{3}$$

$$\frac{\partial(\rho h v)}{\partial t} + \frac{\partial(\rho h v u)}{\partial x} + \frac{\partial}{\partial y}\left(\rho h v v + \frac{1}{2}\rho g h^2\right) = \rho g h S_{0y} - \rho C_f \|u\| v \tag{4}$$

where $g$ is the gravitational acceleration; $S_{0x} = -\frac{\partial(b_0+b)}{\partial x}$ and $S_{0y} = -\frac{\partial(b_0+b)}{\partial y}$ are the $x$ and $y$ components of bed slope; $b_0$ is the elevation of the immobile bed; $b$ is the depth of the mobile bed; $C_f = g n^2 h^{-\frac{1}{3}}$ is the drag coefficient; $n$ is Manning's roughness coefficient; $\|u\| = \sqrt{u^2 + v^2}$ is the magnitude of the flow velocity; $u = \begin{pmatrix} u \\ v \end{pmatrix}$ is the velocity vector. Although Manning's equation was derived for steady flow, this model assumes the instantaneous bed shear stress can be estimated using Manning's equation [16,21,23,24]. Because flow depth and velocity are spatial and temporal variables, the bed shear stress in unsteady flow is also a spatial and temporal variable, changing with flow depth and velocity.

The system of governing equations can also be written in a vectorial form as:

$$\frac{\partial Q}{\partial t} + \frac{\partial F_x(Q)}{\partial x} + \frac{\partial F_y(Q)}{\partial y} = S_0(U) - S_f(U) + S_b(U) \tag{5}$$

where $U$ and $Q$ are the vectors of primitive variables and conservative variables:

$$U = \begin{pmatrix} h \\ \rho \\ u \\ v \end{pmatrix}, Q = \begin{pmatrix} h \\ \rho h \\ \rho h u \\ \rho h v \end{pmatrix} \tag{6}$$

$F_x(Q)$ and $F_y(Q)$ are the vectors of advective fluxes:

$$F_x(Q) = \begin{pmatrix} hu \\ \rho h u \\ \rho h u u + \frac{1}{2}\rho g h^2 \\ \rho h u v \end{pmatrix}, F_y(Q) = \begin{pmatrix} hv \\ \rho h v \\ \rho h v u \\ \rho h v v + \frac{1}{2}\rho g h^2 \end{pmatrix} \tag{7}$$

and $S_0(U)$, $S_f(U)$, and $S_b(U)$ are the bed slope, bed friction, and bed material source terms:

$$S_0(U) = \begin{pmatrix} 0 \\ 0 \\ \rho g h S_{0x} \\ \rho g h S_{0y} \end{pmatrix}, S_f(U) = \begin{pmatrix} 0 \\ 0 \\ \rho C_f \|u\| u \\ \rho C_f \|u\| v \end{pmatrix}, S_b(U) = \begin{pmatrix} S_b \\ \rho_b S_b \\ 0 \\ 0 \end{pmatrix} \tag{8}$$

Equation (5) represents a system of time-dependent nonlinear hyperbolic partial differential equations. This system may result in sharp and discontinuous solutions even if starting from a continuous initial condition.

### 2.1. Nonequilibrium Sediment Transport Model

The evolution of mobile bed is described by the conservation equation of bed material:

$$\frac{\partial b}{\partial t} = -S_b \tag{9}$$

The mass exchange between sediment-laden flow and mobile bed is evaluated by the nonequilibrium sediment transport equation [10,25]. It reads:

$$S_b = \frac{1}{1-\phi}\frac{(q_b - q_b^*)}{L} \tag{10}$$

where $L$ is the nonequilibrium adaption length of sediment transport; $q_b$ is the actual flux of sediment transport; and $q_b^*$ is the sediment transport capacity or the equilibrium sediment

transport rate. Based on the recommendations of [26,27], the sediment transport capacity is calculated by the modified Meyer–Peter–Müller (MPM) formula:

$$q_b^* = 12\sqrt{sgd_{50}^3}\left(\theta - \theta_c\right)^{1.5} \tag{11}$$

where $\theta = \frac{u_*^2}{sgd}$ is the Shields number; $u_* = C_f^{\frac{1}{2}}\|u\|$ is the friction velocity; $\theta_c$ is the critical Shields number.

The nonequilibrium adaptation length in Equation (10) denotes the spatial lag between actual sediment transport rate and its saturation and equilibrium rate. It is calculated by [23]:

$$L = max\left(L_b, \frac{h\|u\|}{\alpha_0\omega_0}\right) \tag{12}$$

where $L_b$ is the adaption length of bedload; $\alpha_0$ is the adaption coefficient of suspended-load; and $\omega_0$ is the settling velocity of a single sediment particle. Since there has been no consensus on $L_b$ and $\alpha_0$, both parameters are treated as calibration parameters in the proposed model. The settling velocity of sediment particle is calculated by [28]:

$$\omega_0 = \sqrt{\left(\frac{13.95\nu}{d_{50}}\right)^2 + 1.09sgd_{50}} - \frac{13.95\nu}{d_{50}} \tag{13}$$

where $\nu$ is the kinematic viscosity of water; $s = \frac{\rho_s}{\rho_w} - 1$ is the specific gravity of submerged sediment particle; and $d_{50}$ is the median diameter of sediment particle.

*2.2. Numerical Methods*

2.2.1. Godunov-Type Finite Volume Method

The integral form of the VDSWEs over a computational cell can be written as:

$$\frac{\partial}{\partial t}\int_\Omega Q d\Omega + \int_{\partial\Omega}(Fn)d\partial\Omega = \int_\Omega S_0 d\Omega - \int_\Omega S_f d\Omega + \int_\Omega S_b d\Omega \tag{14}$$

where $\Omega$ represents a computational cell; $\partial\Omega$ is the boundary of the cell; $F = \begin{pmatrix} F_x & F_y \end{pmatrix}$ is the matrix of fluxes; $n = \begin{pmatrix} n_x \\ n_y \end{pmatrix}$ is the outward unit normal vector; $n_x$ and $n_y$ are the components of the normal vector in $x$ and $y$ directions, respectively; and $Fn = F_x n_x + F_y n_y$ are the fluxes normal to the cell boundary designated by $n$. In this study, the fully coupled system of governing equations is solved by the first-order Godunov-type finite volume method over a regular Cartesian mesh. For a rectangular cell, the fluxes across the cell boundaries are calculated by:

$$\int_{\partial\Omega}(Fn)d\partial\Omega = \sum_{k=1}^4 F_k(Q_L, Q_R)n_k l_k \tag{15}$$

where subscript $k$ is the edge index; subscript $L$ and $R$ denote the left and right edges of a cell boundary, respectively; $F_k(Q_L, Q_R)$ is the Riemann fluxes at the edge $k$; $Q_L$ and $Q_R$ are the vectors of conservative variables reconstructed at the left and right edges, respectively; $n_k$ is the unit normal vector for the $k$th edge; $l_k$ is the length of the $k$th edge.

A first-order, two-step fractional scheme is employed in the numerical model to update the solution at each time step:

Step one:

$$Q^{(*)} = Q^{(n)} - \frac{\Delta t}{\Delta A}\sum_{k=1}^4 F_k\left(Q_L^{(n)}, Q_R^{(n)}\right)n_k l_k + \Delta t\left(S_0^{(n)} - S_f^{(n)}\right) \tag{16}$$

Step two:

$$
\begin{cases}
Q^{(n+1)} = Q^{(*)} + \Delta t S_b^{(*)} \\
b^{(n+1)} = b^{(n)} - \Delta t \frac{S_b^{(*)}}{1-\phi}
\end{cases}
\tag{17}
$$

where superscript $(n)$ denotes the solution at time step $n$; superscript $(n+1)$ denotes the time step $(n+1)$; superscript $(*)$ denotes the intermediate solution between time step $n$ and $n+1$.

Time steps are limited by two stability conditions. The first is the Courant–Friedrichs–Lewy (CFL) condition that requires the maximum CFL number should not be greater than 0.5 at each time step [29]. The second is the bed change condition that limits the change in mobile bed at each time step up to 10% of the local bed depth [9].

### 2.2.2. HLLC Approximated Riemann Solver

The HLLC (Harten–Lax–van Leer–Contact) approximate Riemann solver [15] is extended to calculate numerical fluxes across cell boundaries. The wave structure of a typical HLLC solution is demonstrated in [21]. The solution is separated by three waves: the speeds of the left wave ($S_L$), the middle wave ($S_*$), and the right wave ($S_R$), which are defined later using Equations (20) and (23). There are four constant states in the solution: the left state ($Q_L$), the right state ($Q_R$), the left star state ($Q_{*L}$), and the right star state ($Q_{*R}$). The corresponding HLLC numerical flux $F_{HLLC}(Q_L, Q_R)$ is defined as:

$$
F_{HLLC}(Q_L, Q_R) = \begin{cases}
F_L & 0 \le S_L \\
F_{*L} & S_L < 0 \le S_* \\
F_{*R} & S_* < 0 \le S_R \\
F_R & S_R < 0
\end{cases}
\tag{18}
$$

where $F_L = F(Q_L)$ represents the supercritical flow from the left to the right; $F_R = F(Q_R)$ represents the supercritical flow from the right to the left; $F_{*L}$ and $F_{*R}$ are the fluxes at the star regions. By applying the Rankine–Hugoniot conditions across the waves, $S_L$ and $S_R$, the numerical fluxes at the star regions can be calculated by:

$$
\begin{cases}
F_{*L} = F_L + S_L(Q_{*L} - Q_L) \\
F_{*R} = F_R + S_R(Q_{*R} - Q_R)
\end{cases}
\tag{19}
$$

The middle wave speed is estimated by [21]:

$$
S_* = \frac{p_L - p_R + \rho_R h_R U_R(S_R - U_R) + \rho_L h_L U_L(U_L - S_L)}{\rho_R h_R(S_R - U_R) + \rho_L h_L(U_L - S_L)}
\tag{20}
$$

where $p_L = \frac{1}{2}\rho_L g h_L^2$ and $p_R = \frac{1}{2}\rho_R g h_R^2$ are the hydrostatic pressures at the left and right edges, respectively; $U_L = u_L \cdot n$ and $U_R = u_R \cdot n$ are the normal velocities at the left and right edges, respectively; $\rho_L$ and $\rho_R$ are the densities at the left and right, respectively. The left and right states at the star regions can be determined by:

$$
Q_{*L} = h_L \frac{U_L - S_L}{S_* - S_L}
\begin{pmatrix}
1 \\
\rho_L \\
\rho_L(S_* n_x + u_L n_y) \\
\rho_L(v_L n_x + S_* n_y)
\end{pmatrix}
\tag{21}
$$

$$
Q_{*R} = h_R \frac{U_R - S_R}{S_* - S_R}
\begin{pmatrix}
1 \\
\rho_R \\
\rho_R(S_* n_x + u_R n_y) \\
\rho_R(v_R n_x + S_* n_y)
\end{pmatrix}
\tag{22}
$$

The left and right wave speeds are estimated by:

$$
\begin{cases}
S_L = min(U_L - \sqrt{gh_L}, \tilde{u} - \sqrt{g\tilde{h}}) \\
S_R = max(U_R - \sqrt{gh_R}, \tilde{u} + \sqrt{g\tilde{h}})
\end{cases}
\tag{23}
$$

where $\tilde{h} = \frac{h_L + h_R}{2}$ and $\tilde{u} = \frac{U_R\sqrt{h_R} + U_L\sqrt{h_L}}{\sqrt{h_R} + \sqrt{h_L}}$ are the Roe-averaged flow depth and velocity.

### 2.2.3. Well-Balanced Scheme for the VDSWEs

To illustrate the method, this section used one-dimensional equations simplified from SWEs. For a stationary flow ($u = 0$), the momentum equation of SWEs becomes:

$$
\frac{\partial}{\partial x}\left(\frac{1}{2}gh^2\right) = ghS_{0x}
\tag{24}
$$

If a numerical scheme can maintain this stationary state, the scheme is called a well-balanced scheme [30], and it is said to satisfy the conservation property (or *C*-property) [22].

As to the VDSWEs, the momentum equation including the variable-density term changes into:

$$
\frac{\partial}{\partial x}\left(\frac{1}{2}\rho gh^2\right) = \rho ghS_{0x}
\tag{25}
$$

It is obvious that the definition of a well-balanced scheme or the conservative property can be extended to the VDSWEs, if a numerical scheme can maintain Equation (25). Rearranging Equation (25) into a similar form as Equation (24):

$$
\frac{\partial}{\partial x}\left(\frac{1}{2}gh^2\right) = ghS_{0x} - \frac{gh^2}{2\rho}\frac{\partial \rho}{\partial x}
\tag{26}
$$

Since the density gradient term on the right-hand side comes from the advection term of Equation (25), the term bears an upwind characteristic. It is difficult to maintain the conservative property of a numerical scheme if this term is discretized using a finite difference scheme (e.g., a central difference method) in [1].

To create a well-balanced scheme, the hydrostatic reconstruction approach [31] for the SWEs is incorporated into the proposed model. According to this approach, bed elevations at the cell interface $(i + \frac{1}{2}, j)$ between the cell $(i, j)$ and the cell $(i + 1, j)$ are evaluated by:

$$
B_{i+\frac{1}{2},j} = max(B_{i,j}, B_{i+1,j})
\tag{27}
$$

where $B = b + b_0$ is bed elevation; subscript $i$ and $j$ are the cell indices in $x$ and $y$ directions; and subscript $1/2$ represents the cell boundary. Following this, flow depths at the left and right sides of a cell boundary are reconstructed as:

$$
\begin{cases}
h_L = max(0, h_{i,j} + B_{i,j} - B_{i+\frac{1}{2},j}) \\
h_R = max(0, h_{i+1,j} + B_{i+1,j} - B_{i+\frac{1}{2},j})
\end{cases}
\tag{28}
$$

The reconstructed conservative variables at the cell boundary for 2D VDSWEs are equal to:

$$
\boldsymbol{Q}_L = \begin{pmatrix} h_L \\ \rho_{i,j}h_L \\ \rho_{i,j}h_L u_{i,j} \\ \rho_{i,j}h_L v_{i,j} \end{pmatrix}, \boldsymbol{Q}_R = \begin{pmatrix} h_R \\ \rho_{i+1,j}h_R \\ \rho_{i+1,j}h_R u_{i+1,j} \\ \rho_{i+1,j}h_R v_{i+1,j} \end{pmatrix}
\tag{29}
$$

Accordingly, the bed slope term at the cell center is discretized for 2D VDSWEs as:

$$S_0 = \begin{pmatrix} 0 \\ 0 \\ \dfrac{\rho_{i,j} g \left( h_{E,L}^2 - h_{W,R}^2 \right)}{2\Delta x} \\ \dfrac{\rho_{i,j} g \left( h_{N,L}^2 - h_{S,R}^2 \right)}{2\Delta y} \end{pmatrix} \tag{30}$$

where subscript $E$ ($i+1$), $W$ ($i$), $N$ ($j+1$), and $S$ ($j$) represent the east, west, north, and south sides of a cell. For one-dimensional flow, the fourth component in Equations (29) and (30) is neglected.

To prove the well-balanced property of the scheme, the numerical discretization to a stationary flow is presented below. For a flow in hydrostatic equilibrium state, the wave speeds are:

$$\begin{cases} S_L = min\left( -\sqrt{gh_L}, -\sqrt{g\tilde{h}} \right) < 0 \\ S_R = max\left( +\sqrt{gh_R}, +\sqrt{g\tilde{h}} \right) > 0 \\ S_* = 0 \end{cases} \tag{31}$$

For the cell $i$, the left-hand side (LHS) of Equation (25) is discretized as:

$$\text{LHS} = \frac{F_{i+\frac{1}{2}} - F_{i-\frac{1}{2}}}{\Delta x} = \frac{\frac{\rho_i g h_L^2}{2} - \frac{\rho_i g h_R^2}{2}}{\Delta x} = \frac{\rho_i g \left( h_L^2 - h_R^2 \right)}{2\Delta x} \tag{32}$$

while the right-hand side (RHS) of Equation (25) equals:

$$\text{RHS} = \frac{\rho_i g \left( h_L^2 - h_R^2 \right)}{2\Delta x} \tag{33}$$

Since the left-hand side is exactly equal to the right-hand side, the proposed numerical scheme is well-balanced. This contrasts to the unbalanced scheme that discretizes Equation (26) using a central difference scheme for the density gradient. The left-hand side of Equation (26) is discretized as:

$$\text{LHS} = \frac{F_{i+\frac{1}{2}} - F_{i-\frac{1}{2}}}{\Delta x} = \frac{\frac{g h_L^2}{2} - \frac{g h_R^2}{2}}{\Delta x} = \frac{g \left( h_L^2 - h_R^2 \right)}{2\Delta x} \tag{34}$$

and the right hand side is:

$$\text{RHS} = \frac{g \left( h_L^2 - h_R^2 \right)}{2\Delta x} + \frac{g h_i^2}{2\rho_i} \left( \frac{\rho_{i+\frac{1}{2}} - \rho_{i-\frac{1}{2}}}{\Delta x} \right) \tag{35}$$

Apparently, Equations (34) and (35) are not equal, and the additional term is the density gradient term. This unbalanced discretization can cause numerical dispersion at the discontinuous locations.

## 3. Results

The numerical method was tested using four cases: one is a synthetic case of standing contact discontinuity, two are laboratory experiments of dam-break flow over mobile bed surface, and the last is a field case.

### 3.1. Case 1: Standing Contact-Discontinuity

This synthetic case aims to demonstrate the well-balanced property of the model. The case is one-dimensional stationary fluid in a flume ($L = 500$ m) with a flat bed and

both sides of the flume are solid walls proposed by [21]. The vectors in Equation (6) were reduced to three components, $\boldsymbol{U} = (h, \rho, u)^T$. The initial conditions are given as:

$$\boldsymbol{U}(x,0) = \begin{cases} \boldsymbol{U}_L, x <= 250 \text{ m} \\ \boldsymbol{U}_R, x > 250 \text{ m} \end{cases} \tag{36}$$

with:

$$\boldsymbol{U}_L = \begin{pmatrix} 4.0 \\ 1562.5 \\ 0.0 \end{pmatrix}, \boldsymbol{U}_R = \begin{pmatrix} 5.0 \\ 1000.0 \\ 0.0 \end{pmatrix} \tag{37}$$

For the given initial conditions, the discontinuity at $x_0 = 250$ m is a standing contact-discontinuity. By enforcing the zero-velocity condition, the fluid will remain stationary without dispersion. Therefore, the analytical solution is the fluid at the initial stationary state, assuming no vertical flow, using the depth-averaged two-dimensional model.

The numerical simulation is carried out over a Cartesian mesh with $\Delta x = 1$ m. For the simulation, the time step is $\Delta t = 0.02$ s and the simulation time is $t_{max} = 10$ s. The simulation results, including surface profile, density profile, and velocity profile, are shown in Figure 1. The profiles of the results are identical to the stationary solution of this test. Therefore, the well-balanced property of the numerical model is confirmed by this test case.

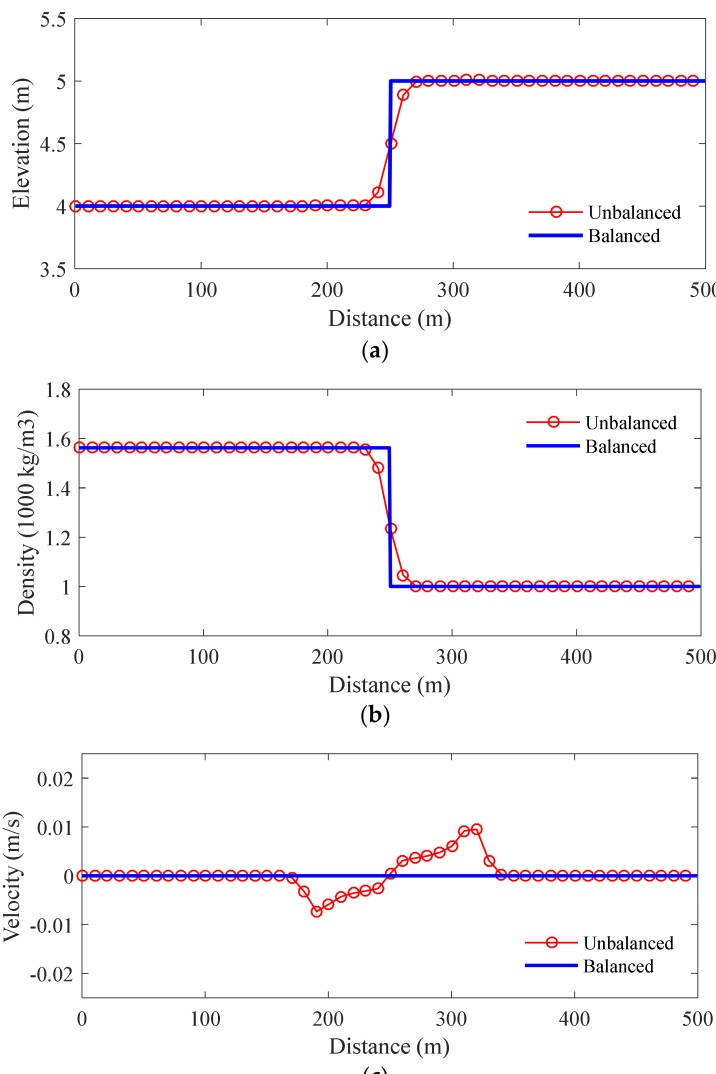

**Figure 1.** Solution of test case 1: (**a**) surface profile; (**b**) density profile; and (**c**) velocity profile.

Besides this, the results of an unbalanced scheme [1] are also plotted in Figure 1. It is apparent that there are two oscillations that propagate toward the upstream and downstream ends of the flume and the unbalanced scheme cannot keep the stationary state solution. These comparisons demonstrate the advantage of a well-balanced scheme over an unbalanced scheme: a well-balanced scheme can maintain the exact equilibrium between the hydrostatic pressure term and the bed slope term and there are no oscillations in the solution of a well-balanced scheme.

### 3.2. Case 2: One-Dimensional Dam-Break Flow over Mobile Bed

In this test case, the model is applied to a one-dimensional laboratory experiment [32]. The objective of this experiment is to investigate the erosional behavior of dam-break flow over mobile beds. The setup of the experiment is shown in Figure 2. The flume is 6 m long, 0.25 m wide, and 0.7 m high. The flume is equipped with a fast downward-moving gate, which is used to simulate idealized dam-break events. The experiment uses a flat, loose granular sediment bed with following parameters: the density of sediment particle, $\rho_s = 2683 \text{kg/m}^3$; the median diameter of sand, $d_{50} = 1.82$ mm; the settling velocity, $\omega_0 = 0.16$ m/s; and the porosity, $\phi = 0.47$. Manning's roughness coefficient was estimated to be $n = 0.0165$ s/m$^{\frac{1}{3}}$.

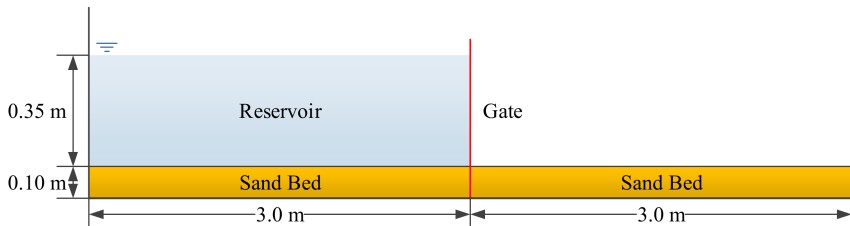

**Figure 2.** Experimental setup of test case 2.

Initially, the reservoir upstream is filled with water to a depth $h = 0.35$ m. The downstream of the flume is dry bed. The mobile bed layer consists of fully saturated sands with an initial thickness $b = 0.1$ m. The upstream boundary condition is a wall boundary. The transmissive boundary condition, which assumes the velocity gradient is zero ($\frac{\partial u}{\partial x} = 0$), is applied to the downstream bounadry. At time $t = 0$ s, the gate was suddenly removed to create a dam-break flow. During the experiment, the dam-break flow was recorded by several high-speed digital cameras. The total experimental time was $t_{max} = 1.5$ s.

The numerical simulation is carried out with the following parameters: $\Delta x = 0.06$ m and $\Delta t = 0.001$ s. In Figure 3, the calculation results are compared with the measured data at T= 0.25 s, 0.50 s, 0.75s, 1.00 s, 1.25 s, and 1.50 s. In addition, the calculated surface and bed elevation in [21] at T = 0.50 s, 1.00 s, and 1.50 s are plotted to compare with the simulated results in this study. The calculated results at T = 0.25 s, 0.75 s, and 1.25 s are not available in [21]. Figure 3 shows reasonable agreement between the calculated results from this study and the measurements. Not only is the propagation of the shock-front accurately predicted, but also the temporal evolution of the free surface agrees well with the interface-tracking results of the digital cameras. Although the bed changes are not significant in the experiment, the magnitudes of the simulated bed changes are consistent with the measurements. However, it is apparent that the results of water surface and bed elevations from [21] deviate more from the measurements compared to the results from this study. At T = 1.00 s and 1.50 s, the simulated water surface and bed elevation from this study are very close to those observed, but the results in [21] over-predicted water surface elevation and bed scour depth. Ref. [21] showed a remarkable dip in both water surface and bed profiles at the gate (x = 0.0 m). This discontinuity of water and bed surfaces at the gate, where both water surface and density are discontinuous at the initial state, is nearly invisible but also seen in this study. This is the location of the maximum of numerical errors regardless of numerical scheme. The results show the well-balanced scheme used in this

study not only improves the simulated results but also minimizes the numerical instability at the discontinuity interface.

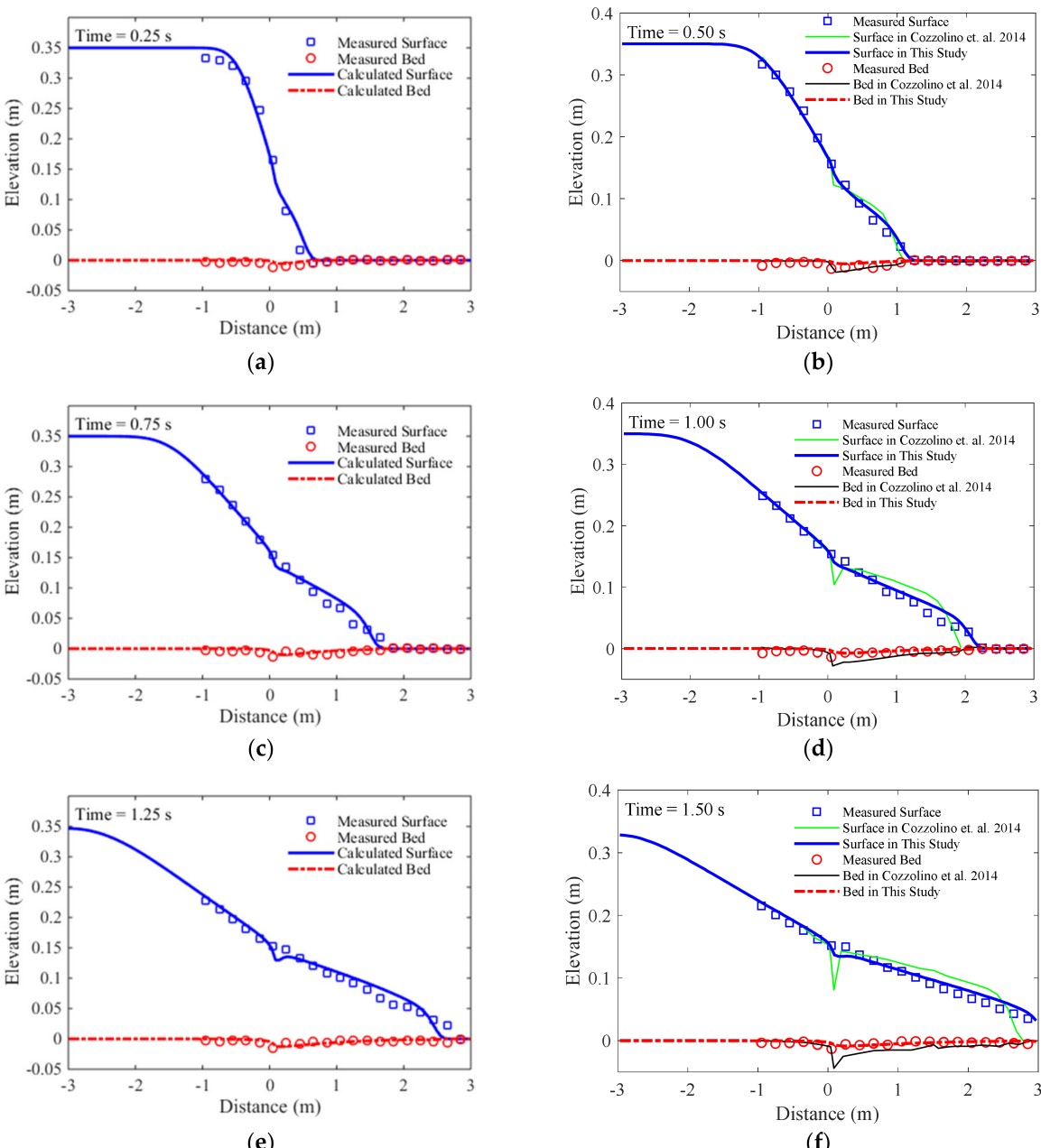

**Figure 3.** Measured and calculated results of test case 2: (**a**) t = 0.25 s; (**b**) t = 0.50 s; (**c**) t = 0.75 s; (**d**) t = 1.00 s (**e**) t = 1.25 s; and (**f**) t = 1.50 s [21].

To quantitatively evaluate the accuracy of the developed model, the root mean square error (RMSE) and the mean relative error (MRE) of the simulated results are calculated using Equation (38) and shown in Table 1.

$$RMSE = \sqrt{\frac{\sum_{i=1}^{n}(C_i - O_i)^2}{n}} \; MRE = \frac{\overline{|C_i - O_i|}}{|O_i|} \qquad (38)$$

where $C_i$ and $O_i$ are calculated and observed values, respectively; n is the total number of observations; and the overbar is the mean value. As shown in Table 1, the RMSEs of the simulated surface profiles are around 0.01 m, while the RMSEs of the simulated bed

profiles are less than 0.005 m. However, the MREs of the simulated bed profiles are much greater than the simulated surface profiles. This reveals that the simulated surface profiles are more accurate than the simulated bed profiles.

**Table 1.** RMSE and MRE of 1D dam-break flow.

| Time (s) | Water Level | | Bed Elevation | |
|---|---|---|---|---|
| | RMSE (cm) | MRE (%) | RMSE (cm) | MRE (%) |
| 0.25 | 1.13 | 3.35 | 0.45 | 23.20 |
| 0.50 | 0.57 | 1.80 | 0.38 | 25.09 |
| 0.75 | 0.78 | 2.79 | 0.39 | 26.47 |
| 1.00 | 0.84 | 3.37 | 0.33 | 22.24 |
| 1.25 | 0.82 | 3.58 | 0.35 | 23.75 |
| 1.50 | 0.87 | 4.83 | 0.32 | 27.35 |
| Average | 0.84 | 3.29 | 0.37 | 24.68 |

The simulated concentration profiles at different time instants are plotted in Figure 4. The existence of high concentration at the shock front (about 20%) justifies the significance of the variable-density effect. This is also a challenge to the numerical modeling of sediment transport because the concentration profile is discontinuous at the shock wave front. Figure 4 demonstrated that the proposed model can capture this concentration discontinuity without introducing spurious oscillations.

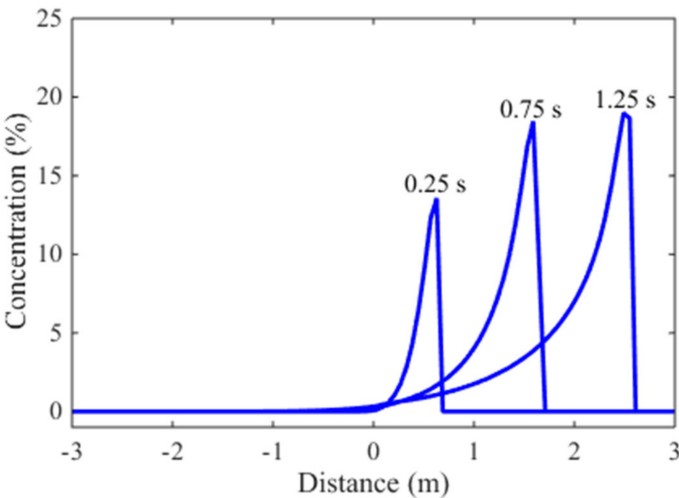

**Figure 4.** Concentration profiles of 1D dam-break flow.

### 3.3. Case 3: Two-Dimensional Dam-Break Flow over Mobile Bed

This laboratory experiment was conducted at the Hydraulics Unit of the Mechanical and Civil Engineering Laboratory, Universite catholique de Louvain, Belgium [33]. The experiment aims to provide a benchmark test case to validate numerical models for the simulation of dam-break flow over a mobile bed [7,9,34]. The experiment is a dam-break flow from an upstream reservoir flowing into a flume over a mobile bed made of uniform coarse particles. A schematic view of the flume is shown in Figure 5. The length of the flume is 36 m; the width of the flume is 3.6 m. The breached dam is in a narrow reach (1 m long and 1 m wide) between two impervious blocks. The upstream of the flume was a reservoir, whereas the downstream is a flooded channel. The origin of the coordinates is situated at the center of the dam. Manning's coefficient for the sandy bed is given as $0.010 \text{ s/m}^{\frac{1}{3}}$.

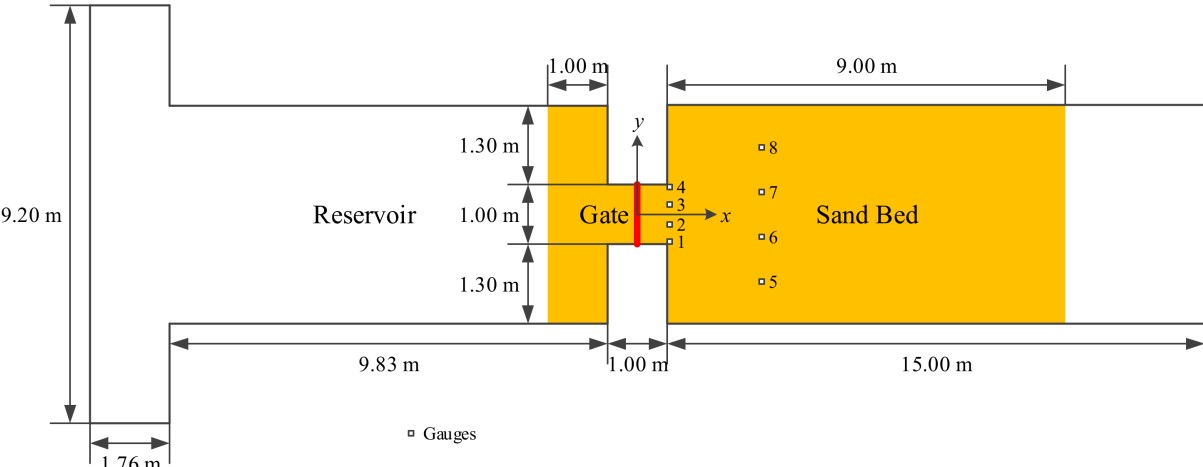

**Figure 5.** Experimental setup of 2D dam-break flow.

The bottom of the flume was covered with a layer of saturated sand extending from 1 m upstream of the dam to 9 m downstream. The thickness of the mobile sand layer was 8.5 cm. The related properties of the sand layer are: the mean diameter of the particles $d_{50} = 1.61$ mm; the porosity of bed material $\phi = 0.42$; the density $\rho_s = 2630$ kg/m$^3$; and the critical Shields parameter $\theta_c = 0.047$. Manning's roughness coefficient for this sandy channel was measured as $n = 0.0165$ s/m$^{\frac{1}{3}}$.

The dam-break flow was triggered by rapidly lifting the gate separating the reservoir and the channel. The initial water level in the reservoir was 0.47 m, while the downstream channel was initially dry (h = 0 m). The wall boundary condition was applied to the upstream end of the reservoir, and similarly, the transmissive boundary condition is imposed at the downstream outlet of the flume. The experiment lasted for 20 s. During the experiment, eight sonic water level gauges were used to record water levels. The bed profiles were measured after the experiment along three longitudinal lines: $y = 0.2$ m, $y = 0.7$ m, and $y = 1.45$ m.

For the numerical simulation, the flume is discretized using a Cartesian mesh ($\Delta x = \Delta y = 0.1$ m), with a total of 10,602 rectangular cells. The time step is set to be $\Delta t = 0.01$ s. The comparisons of simulated and measured water level hydrographs at eight gauges are shown in Figure 6. The quantitative evaluation indices, i.e., the RMSEs and the mean relative errors, are shown in Table 2. The comparisons showed that, at all eight gauges, both the arrival times of the dam-break waves and the peak water levels are accurately predicted. At the early stage of experiment, the water levels at the gauges, G2 and G3, are overpredicted, while the water levels are underpredicted at the gauges, G5 and G8. At the late stage, the water levels are underpredicted at G1, G4, G6, G7, and G8. For the symmetric gauge pairs (G1/G4, G2/G3, G5/G8, and G6/G7), the simulated hydrographs are nearly symmetrical except for the results at G5 and G8 at the late stage. The RMSEs of the predicted water level hydrographs are between 0.015 m and 0.02 m, while the relative errors are between 12% and 22%. The results at the upstream gauges (G1, G2, G3, and G4) are better than the downstream gauges (G5, G6, G7, and G8). A possible explanation is that, as the dam-break flow propagates from upstream to downstream, three-dimensional flow effects become more dominant.

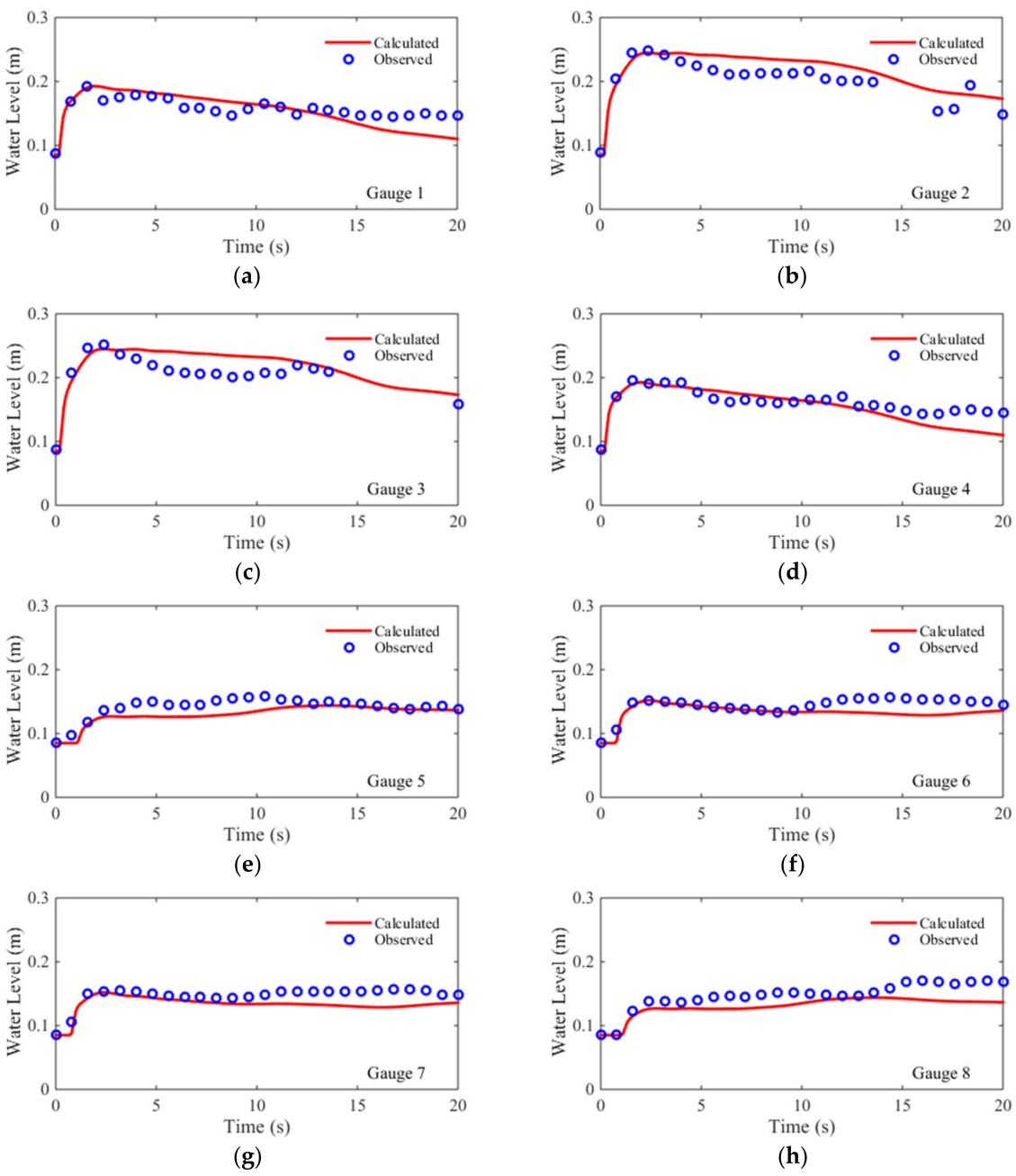

**Figure 6.** Measured and calculated water levels of 2D dam-break flow at: (**a**) Gauge 1; (**b**) Gauge 2; (**c**) Gauge 3; (**d**) Gauge 4; (**e**) Gauge 5; (**f**) Gauge 6; (**g**) Gauge 7; and (**h**) Gauge 8.

**Table 2.** RMSE and MRE of water levels of 2D dam-break flow.

| Gauge Number | RMSE (cm) | MRE (%) |
| --- | --- | --- |
| 1 | 1.71 | 15.11 |
| 2 | 2.07 | 12.66 |
| 3 | 2.17 | 12.72 |
| 4 | 1.59 | 13.81 |
| 5 | 1.39 | 18.90 |
| 6 | 1.49 | 20.73 |
| 7 | 1.58 | 21.91 |
| 8 | 1.98 | 21.50 |
| Average | 1.75 | 17.17 |

The comparisons between the calculated and measured bed profiles along the longitudinal lines at the end of the experiment are shown in Figure 7. The RMSEs and the mean relative errors are shown in Table 3. The magnitudes of RMSEs are between 0.009 m and 0.017 m, while the relative errors are between 13% and 33%. The simulated mean bed profiles matched the measurements at all three measured longitudinal lines. Although the calculated and the measured bed profiles have the same trends, the calculated bed profiles are smoother than the measured bed profiles. This implies that the developed model cannot capture the local perturbations generated by 3D flow effects [34]. These perturbations are caused by the flow in the vertical direction and cannot be captured by a 2D depth-averaged model. On the other hand, the energy dissipations caused by these perturbations can be calculated by a turbulence modeling method [24]. It thus follows that further research is necessary for incorporating a turbulence model into the proposed model.

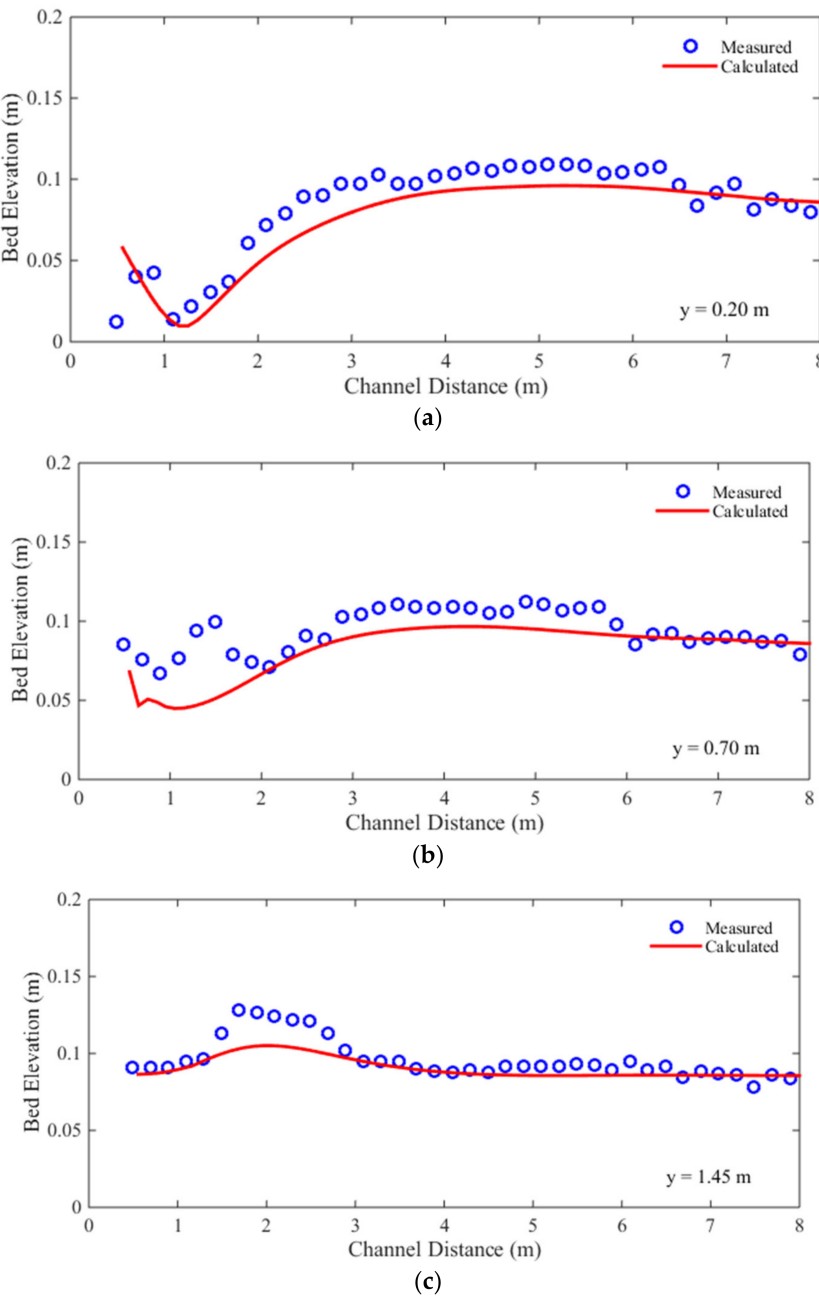

**Figure 7.** Measured and calculated bed profiles of 2D dam-break flow: (**a**) *y* = 0.20 m; (**b**) *y* = 0.70 m; (**c**) *y* = 1.45 m.

**Table 3.** RMSE and MRE of bed elevations of 2D dam-break flow.

| Bed Profile (m) | RMSE (cm) | MRE (%) |
| --- | --- | --- |
| 0.20 | 1.24 | 13.03 |
| 0.70 | 1.67 | 32.65 |
| 1.45 | 0.91 | 17.20 |
| Average | 1.27 | 20.96 |

*3.4. Case 4: 1996 Lake Ha! Ha! Catastrophic Flood Event*

In this case, the performance of the proposed model is tested against a field event: the 1996 Lake Ha! Ha! catastrophic flood event, in the Saguenay region of Quebec, Canada (Figure 8). A detailed description of this flood event and extensively documented data are provided by [35,36]. Ref. [35] gave a detailed geomorphic description of the event. Refs. [26,37,38] carried out a one-dimensional numerical simulation of this event.

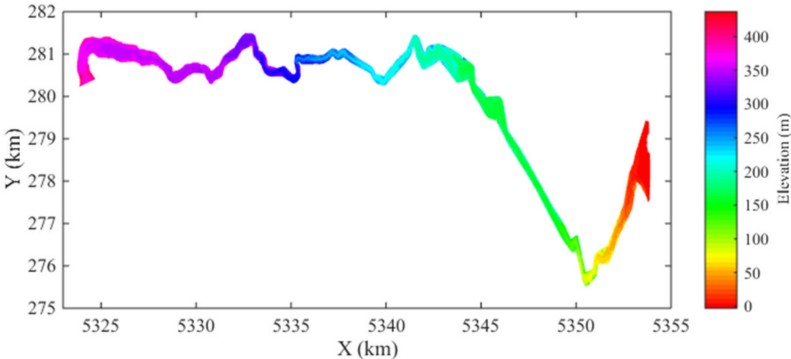

**Figure 8.** DEM map of the Ha! Ha! River, Canada.

From 18 to 21 July 1996, an extreme precipitation event affected the Saguenay region of Quebec, Canada. At the Ha! Ha! Lake, an earth-fill dike was being overtopped by up to 0.26 m of water, and a new outlet channel formed. Overall, $59 \times 10^6$ m$^3$ of water was estimated to have drained from the lake. The failure of the dike resulted in a peak discharge of 8 times the 100-year flood. The Ha! Ha! River was severely damaged by the resulting flood flow [35].

The numerical simulation started with the digital elevation model (DEM) of the Ha! Ha! River, which was surveyed in May 1994 [36]. The spatial data are based on the Modified Transverse Mercator (MTM) projection, zone 7 coordinates (NAD83). The spatial ranges of the DEM data are: 5,318,000 m $\leq x \leq$ 5,354,000 m on the east–west direction, and 275,000 m $\leq y \leq$ 282,000 m in the north–south direction. In addition to the DEM data, the geometry data of evenly spaced cross sections are also provided. These 363 cross sections are spaced at 100 m intervals and oriented approximately normal to the central line of the Ha! Ha! River. Each cross section is identified by its streamwise distance measured from the failed dyke. Despite abundant data, it must be pointed out that the average error of the preflood riverbed elevations is estimated to be about 2 m [26].

Based on the field observation and photo interpretation, superficial materials exposed along the channel and floodplains are classified as non-cohesive sediment (sand and gravel), cohesive sediment (glaciomarine or glaciodiamicton), or bedrock [36]. Due to the lack of sufficient field data, the influence of cohesion on sediment transport was not considered in the numerical model. According to [26], the median diameter of bed material for the whole Ha! Ha! River is equal to 0.5 mm, the density of bed material is $\rho_s = 2650$ kg/m$^3$ and the porosity equals to 0.4. Besides this, the influence of bedrock on the channel profile was studied in [36], the bedrock constraint was also imposed in this simulation for which the outcrops of bedrocks or coarse glacial deposits are assumed to act as rigid, non-erodible beds.

Initially, the Ha! Ha! River is assumed to be dry everywhere. The discharge hydrograph shown in Figure 9 is used as the upstream inflow boundary condition, and the inflow

is assumed to be clear water [26]. The downstream boundary at the Ha! Ha! Bay is set to satisfy the transmissive boundary condition. The overtopping occurred at 14:00 on 19 July 1996. The lake is estimated to be emptied in 18 h. During the event, the water level of the lake dropped from 381 m to 370 m. The total simulation time is set to be 24 h. For the numerical simulation, the river is discretized by a Cartesian mesh with $\Delta x = \Delta y = 20$ m. The time step size is $\Delta t = 0.5$ s.

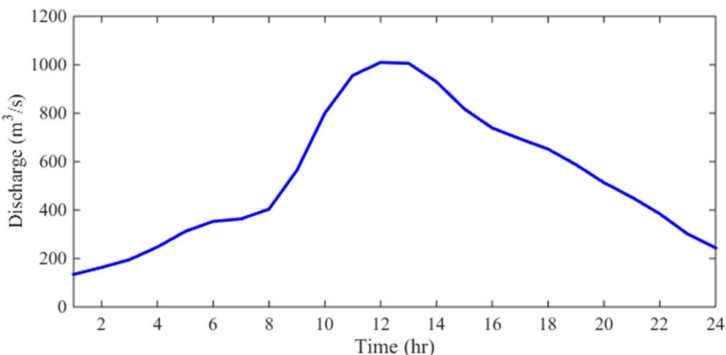

**Figure 9.** Inlet discharge hydrograph of 1996 Lake Ha! Ha! Flood event.

Comparisons of the selected cross sections along the river are shown in Figure 10. The cross-sectional changes are reasonably well predicted by the model. The RMSEs and MREs of bed elevation changes are calculated for all cross sections and are shown in Figure 11. The average of RMSEs is 4.4 m, and the average of MREs is 49.53%. As shown in Figure 12, among all the data, 50% of the RMSEs are less than 3.3 m, and 90% of the RMSEs are less than 7.9 m. As to the MREs, about 50% of cross sections have MRE values less than 40%, and 90% of the MREs are less than 70%.

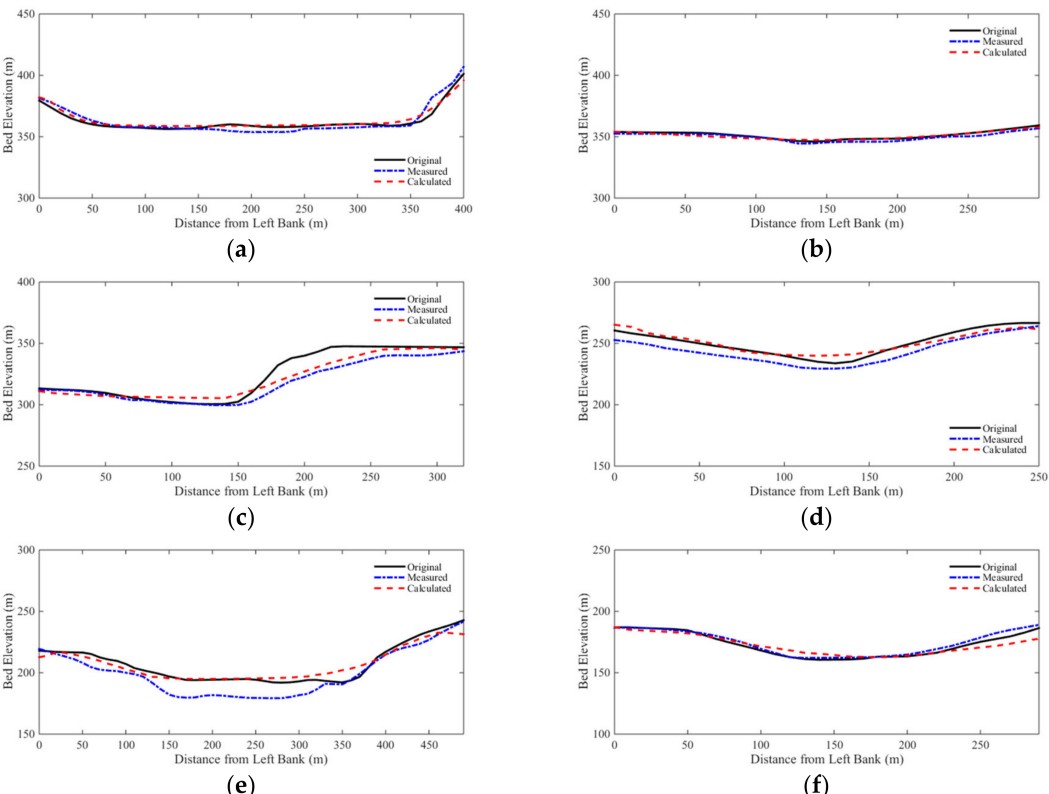

**Figure 10.** *Cont.*

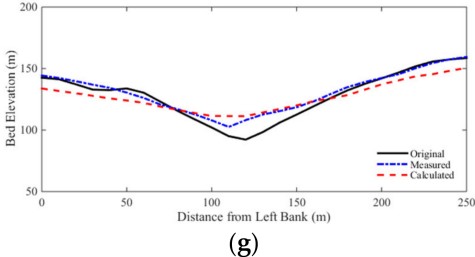

(**g**)

**Figure 10.** Measured and calculated bed cross sections of Lake Ha! Ha! flood event: (**a**) cross section #20; (**b**) cross section #70; (**c**) cross section #120; (**d**) cross section #170; (**e**) cross section #220; (**f**) cross section #270; and (**g**) cross section #320.

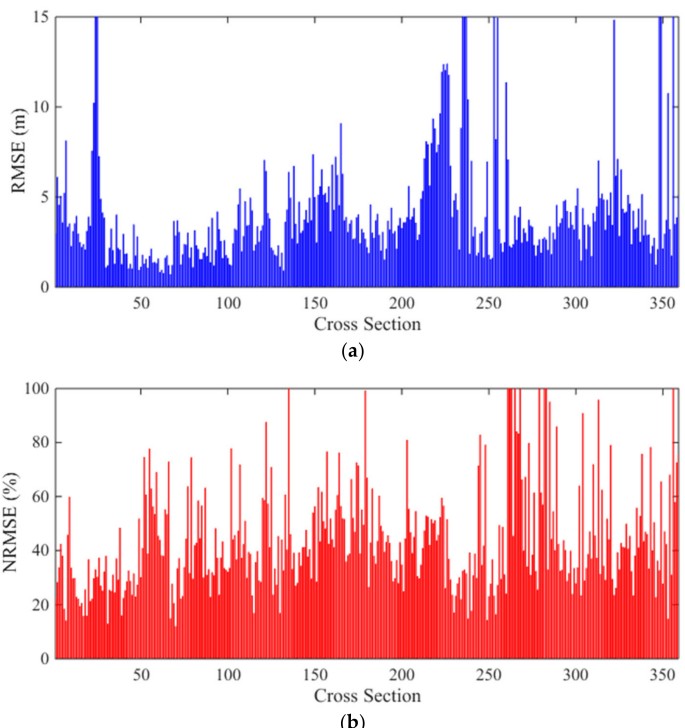

**Figure 11.** Analysis of calculated bed cross sections of Lake Ha! Ha! flood event: (**a**) RMSEs; (**b**) MREs.

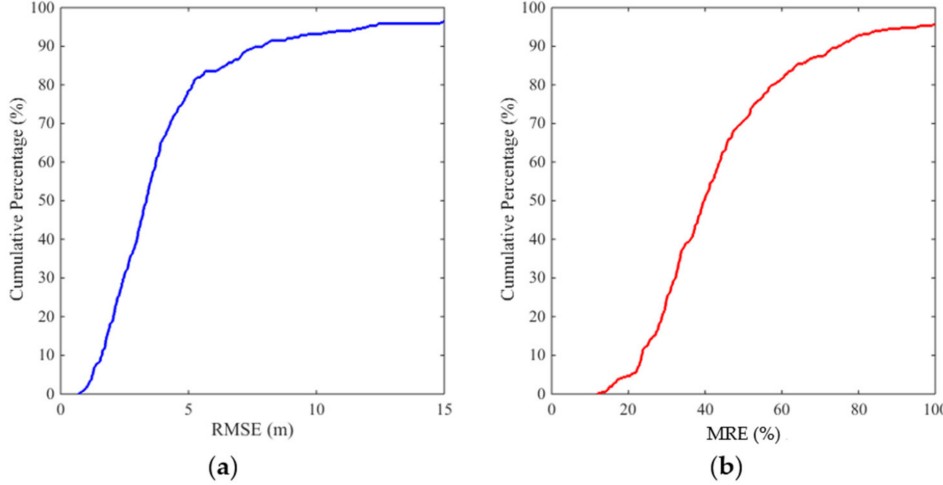

**Figure 12.** Cumulative percentage distributions of Lake Ha! Ha! flood event: (**a**) RMSE; and (**b**) MRE.

In the longitudinal direction, the measured and calculated thalwegs are plotted in Figure 13. The RMSE of thalweg change is 5.3 m, and the MRE is 20.18%. It is obvious that the model performed better in the longitudinal direction than in the cross-sectional direction. Additionally, the simulated thalweg profile at the middle reach is overestimated, especially at 23 km from the channel mouth. According to [26], the outcrops of bedrocks control this part of the channel. During the 1996 flood event, a new reach formed at the right floodplain and the bed was eroded up to 20 m there. The proposed model failed to predict this geomorphic change.

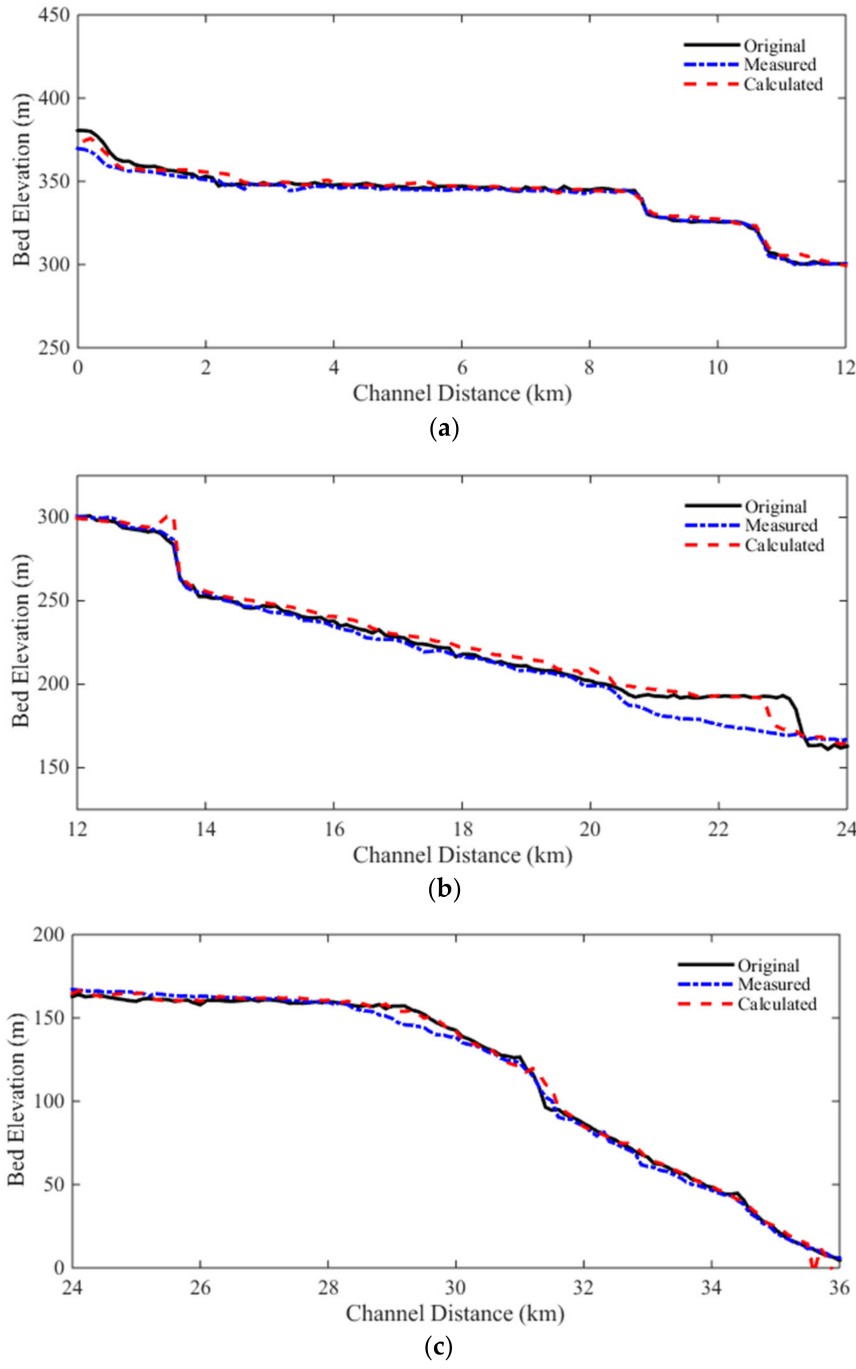

**Figure 13.** Measured and calculated thalwegs of Lake Ha! Ha! flood event: (**a**) 0–12 km; (**b**) 12–24 km; (**c**) 24–36 km.

For this real-world flood event, although the RMSEs of bed elevation (in meter scale) are two orders of magnitude greater than the laboratory cases (in centimeter scale), the MREs of bed elevation (49.53% in horizontal direction and 20.18% in longitudinal direction) are at the same order of magnitude as the laboratory cases (24.68% for 1D dam-break flow case, 20.96% for 2D dam-break flow case). Since there is no consensus on which sediment transport equation is most suitable for a specific river reach, an appropriate calibration procedure using different bed load transport equations and varying Manning's roughness coefficients is needed to reach better matches of modeling results with observations. The focus of this paper is the new well-balanced numerical scheme for solving the VDSWEs. This calibration procedure was not performed for each testing case. Nevertheless, the simulated results of bed-elevation changes satisfactorily match both the laboratory and field observations. Therefore, the developed well-balanced numerical scheme for solving VDSWEs is proven to be a robust and accurate method for simulating unsteady sediment-laden flows over mobile beds.

## 4. Conclusions

A two-dimensional, well-balanced, nonequilibrium sediment transport model for the simulation of unsteady sediment-laden flows over mobile beds is developed and tested. In comparison with previously developed 2D models, the governing equations are based on the original formulation of the VDSWEs, which preserves the conservative property of sediment-laden flow. In the model, this system of governing equations is solved by a fully coupled first-order Godunov-type finite volume method. The HLLC Riemann solver is extended to the two-dimensional VDSWEs to calculate the Riemann fluxes across cells' boundaries. The model adopts a well-balanced scheme that maintains the exact balance between the momentum term and the bed slope term by incorporating the term derived from sediment concentration into the advective terms in momentum equations.

The performance of the model is verified by four test cases. A synthetic case with an analytic solution is used to verify the well-balanced property of the model. Two laboratory experimental studies of 1D and 2D dam-break flows over mobile beds show the accuracy of the model for reproducing not only bed profiles but also the flood propagation processes. The last case is the 1996 Lake Ha! Ha! flood event. The model's results in bed-cross and thalweg sections reasonably match field observations. The accuracy and simplicity of the proposed model, together with the robust implementation of a well-balanced numerical scheme, make this model suitable for practical hydraulic engineering applications, such as sedimentation due to dam decommissioning or levee or barrier breaches. However, the model employs a total load transport equation that limits its applicability to sand-sized dominant rivers. For mountain streams, where sediment load consists primarily of coarse sand and gravels, the sediment transport equation needs to be broken into two equations: one for suspended load, and the other for bed load, in the manner of the method in [11]. Additionally, a partial implicit scheme can be implemented to improve the computational time. Nevertheless, the well-balanced numerical scheme used in this model has significantly improved the robustness and stability of numerical solutions and the simulated results match better than previous models.

**Author Contributions:** The authors' contributions are summarized as: conceptualization, C.Y., J.G.D. and Y.D.; methodology, C.Y. and J.G.D.; software, C.Y.; validation, C.Y., J.G.D. and Y.D.; formal analysis, C.Y.; investigation, C.Y., J.G.D. and Y.D..; resources, J.G.D.; data curation, C.Y. and J.G.D.; writing—original draft preparation, C.Y.; writing—review and editing, J.G.D. and Y.D.; visualization, C.Y.; supervision, J.G.D.; project administration, J.G.D.; funding acquisition, J.G.D. All authors have read and agreed to the published version of the manuscript.

**Funding:** This research was funded by US National Science Foundation (NSF), grant number EAR-0846523.

**Data Availability Statement:** The computational programs and test cases are available at the University of Arizona Data repository.

**Acknowledgments:** The research is partially funded by NSF to the University of Arizona. The funding support is essential for authors to complete this research.

**Conflicts of Interest:** The authors declare no conflict of interest.

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
