# Peer review of "Numerical Simulation of Sediment Transport in Unsteady Open Channel Flow"

_water, doi:10.3390/w15142576_

Round 1
Reviewer 1 Report
This is very interesting research, however, I suggest the author to consider the following comments;
1- Literature review is not critical. It is better to present the weak and strong points of the existing methods showing the novelty of the present manuscript.
2 - The assumptions of this research need to be presented. The formulations are biased on uniform flow, however, the findings are based on non-equilibrum condition. Please explain.
3- Present the limitations of this research.
4- More physical justification of results are required.
5- Conclusion may be better presented, showing the novelty and privilege of this research in comparison to existing ones.
Reviewer 2 Report
This manuscript describes a numerical scheme aimed at improving numerical simulation of flow and sediment transport in a channel. The scheme is based on existing literature for the equations governing flow and sediment transport. The development is in the numerical formulation, specifically of the bed elevation and slope. So far as I can see, other aspects (especially time integration steps) call on existing procedures in the literature.
A comparison is made with a previous “unbalanced” model from the literature in the case of a “standing contact-discontinuity” which is unrealistic in that the depth-integrated models used in the comparison (as developed here and from the literature) should give a static state whereas the actual behaviour in the “real world” would not be static. In the other cases of laboratory experiments and a 1996 Lake Ha! Ha! catastrophic flood event, there is comparison (and reasonable agreement) with measurements but not with any previous “state of the art” model to judge whether this development improves results. Overall I think there needs to be more convincing evidence that this development, which is fairly minor, actually leads to significantly improved simulations.
I have not checked the mathematics.
The manuscript needs improvement in presentation, and completion. Specific “Detailed comments” as below need to be addressed.
Detailed comments
The use of English would benefit from improvement (e.g. line 1 ”presented” –> “presents”; line 29 “is” –> “are”) but I generally limit comments to where the meaning is not clear.
The formatting of equations and symbols using “equations” (I guess), in the manuscript for review, has all the equations and symbols displaced upwards relative to the ordinary text. Equation numbers should preferably be aligned at the right-hand side.
Line 42. “Being different from” –> “In contrast with”
Line 138. In the formula I think “2” should be superscript.
Lines 176 and 180. The descriptions of SL, S* and SR in line 176 do not give them values to use in (18). Do you mean wave velocity?
Line 199. Omit “Without any loss of generality,” – not true. Moreover, two-dimensional equations are considered in this section from line 217.
Line 224. No verb. “. . boundary are equal to:”?
Lines 247-248. Now this is in one dimension. This needs stating because U is only a three-component vector in contrast with its definition in equation (6).
Line 252. “the hydrostatic pressure is constant through the flume”. This is only true in a depth-average (depth-integrated) sense. This should be made clear. In reality this initial condition would lead to strong motion.
Lines 297-298. I missed definition of RMSE and NRMSE. What is NRMSE normalised by (especially important in section 3.4)?
Line 300. “0.05” –> “0.005”.
Figure 3. There appears to be a developing problem with the surface elevation calculation around x=0. This needs comment.
Line 328. “which expanded” –> “extending”
Lines 337 and 416. “the transmissive boundary condition” should be given explicitly.
Line 356. “propagating” –> “propagates”
Line 363. “14” –> “13”
Figure 8 seems to be rotated by 90° relative to the DEM description in lines 395-396.
Line 397. “. . sections are spaced”
The sections on Author Contributions, Funding, Data Availability and Conflicts of Interest, and perhaps Supplementary Materials, still need to be written. “Appendix A” and “Appendix B” sections need to be deleted.
The use of English would benefit from improvement (e.g. line 1 ”presented” –> “presents”; line 29 “is” –> “are”) but I generally limit comments to where the meaning is not clear.
Round 2
Reviewer 1 Report
The authors replied to most of the comments.
If the authors say that assumptions are based on unsteady and non-uniform flow, how do they show unsteady and non-uniform flow conditions in estimation of shear velocity and Shields number? In fact no parameter is presented to show the effect of unsteady and non-uniform flow conditions in these estimations. Please clarify.
Reviewer 2 Report
As before, this manuscript describes a numerical scheme aimed at improving numerical simulation of flow and sediment transport in a channel. The scheme is based on existing literature for the equations governing flow and sediment transport. The development is in the numerical formulation, specifically of the bed elevation and slope. So far as I can see, other aspects (especially time integration steps) call on existing procedures in the literature.
A comparison is made with a previous “unbalanced” model from the literature in the case of a “standing contact-discontinuity” which is unrealistic in that the depth-integrated models used in the comparison (as developed here and from the literature) should give a static state whereas the actual behaviour in the “real world” would not be static. In one case of laboratory experiments there is now comparison with a previous model and this is a distinct improvement on the first version of the manuscript. In the other cases of laboratory experiments and a 1996 Lake Ha! Ha! catastrophic flood event, there is comparison (and reasonable agreement) with measurements but not with any previous model to judge whether this development improves results. Overall I think there is now moderately convincing evidence that this development, which is fairly minor, actually leads to significantly improved simulations.
I have not checked the mathematics. However, there is confusion about the use of “NRMSE”; see “Detailed comments”.
Detailed comments
Lines 169, 170. These equations should be numbered (3), (4).
Line 625. “In Cao et al. (2004).” Is not a sentence, what does it refer to?
Lines 753-754. While this is true in a depth-average (depth-integrated) sense, in reality this initial condition would lead to strong motion. I think this distinction should be made clear. Only the depth-integrated characteristics of the equations are being tested.
Line 886 (Equation 37). The formula for NRMSE appears incomplete (no presence of Root Mean Square and denominator needs to be absolute value). Cl, Ol are not defined. Check also line 985 and Tables 2, 3.
Figure 3 panels b, d, f. Spelling of “Surface” in legend is wrong for “This Study”.
Line 928. “which expanded” –> “extending”
Line 1029. “. . and 275,000 m . .”
Lines 1162-1163. I guess you can omit “For research . . . should be used”
Lines 1170-1172 I guess you can omit “Please . . . reported.”
The manuscript would benefit from one more scan for best use of English.
